# Effects of Cellulase and Xylanase Addition on Fermentation Quality, Aerobic Stability, and Bacteria Composition of Low Water-Soluble Carbohydrates Oat Silage

Wei Liu [1,2,3], Qiang Si [1,2,3], Lin Sun [4], Zhijun Wang [1,2,3], Mingjian Liu [1,2,3], Shuai Du [1,2,3], Gentu Ge [1,2,3] and Yushan Jia [1,2,3],*

1   Key Laboratory of Forage Cultivation, Processing and High Efficient Utilization of Ministry of Agriculture and Rural Affairs, Inner Mongolia Agricultural University, Hohhot 010019, China; liuwei0416996@126.com (W.L.); siqiang_nm@126.com (Q.S.); zhijunwang321@126.com (Z.W.); liumj_nm@163.com (M.L.); dushuai_nm@sina.com (S.D.); gegentu@163.com (G.G.)
2   Key Laboratory of Grassland Resources, Ministry of Education, Inner Mongolia Agricultural University, Hohhot 010019, China
3   College of Grassland, Resources and Environment, Inner Mongolia Agricultural University, Hohhot 010019, China
4   Inner Mongolia Academy of Agricultural and Animal Husbandry Sciences, Hohhot 010031, China; sunlin2013@126.com
*   Correspondence: jys_nm@sina.com; Tel.: +86-13087102501

**Abstract:** Most oat forage has low water-soluble carbohydrates (WSC), which may be the main limited factor for silage fermentation safely, but oat is rich in cellulose and hemicellulose; therefore, we assume that xylanase and cellulase as additives can reduce the content of cellulose and xylan in oat silage, increase the microbial fermentable sugar content, and improve the fermentation quality of the silage. After wilting, oats were treated as follows: (i) distributed water (CK); (ii) silages inoculated with xylanase (X); (iii) silages inoculated with cellulase (C), ensiling for 3 days (early stage of silage) and 60 days (late stage of silage), respectively, after ensiling 60 days for a 5-day aerobic exposure study. The pH, neutral detergent fiber (NDF), and acid detergent fiber (ADF) were significantly reduced by xylanase and cellulase treatment during the late stage of silage, and the concentration of lactic acid, acetic acid, and ammonia nitrogen increased remarkably. The WSC content reached its peak with xylanase treatment during the late stage of silage. The content of crude protein (CP) was not affected by additives but by the silage period; CP and ether extract (EE) significantly increased during the late stage of silage compared to the early stage. After ensiling, the bacterial community showed that xylanase and cellulase treatment increased the relative abundance of lactic acid bacteria. *Lactobacillus* has a higher relative abundance with cellulase treatment after 60 days of ensiling; this can effectively reduce the pH of silage and ensure long-term, stable storage of silage. Cellulase and xylanase increased bacterial diversity during aerobic exposure and improved the aerobic stability of silage significantly. This study indicated that different additives and silage periods had significant effects on chemical compositions, fermentation quality, and bacterial community; meanwhile, both additives improved the aerobic stability of silage. In summary, when the WSC of oat is low, cellulase and xylanase have good effects as silage additives, and the comprehensive effect of cellulase is more prominent.

**Keywords:** oat silage; fermentation; aerobic stability; bacteria community; water-soluble carbohydrates

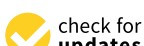



## 1. Introduction

Forage is rich in polysaccharides; cellulose (40 to 45% dry weight in the plant cell wall) exists in both the primary and secondary cell walls and is the most abundant polysaccharide in plant biomass [1]. Xylan, one of the hemicelluloses, is mostly found in secondary walls and is the second most abundant polysaccharide in plant matter [2]. This polysaccharide

cannot be fully utilized in the rumen; rumen microbial can provide about 70% of the metabolizable energy of feedstuffs by fermentation to ruminants, which means that even if the rumen function is in the best state, nearly one-third of the valuable fiber energy is still unused, resulting in waste [3]. However, in low WSC silage materials, polysaccharides also provide the possibility of increasing microbial fermentation substrates [4]. However, most lactic acid bacteria cannot directly ferment and utilize polysaccharides; fibrotic enzymes in silage have been widely used to improve the concentration of fermentation substrate [5]. The enzymes that can be described can cleave the main chains of cellulose and the backbone and branches of hemicellulose and pectin to improve the content of fermentable sugars [6]. In addition, the degradation of oligosaccharides and polysaccharides is slower than starch in the rumen, and the ruminal pH is protected by its fermentation [7].

Oat (*Avena sativa* L.) is a typical grain and forage plant; it is one of the most important feed resources in the world. Ensiling is a common measure for resaving fresh forages and against rotting; the production is less dependent on weather and improves the palatability and storage period [8]. However, during the ensiling process, sugar gradually decreases due to microbial activity, and about 70 g kg$^{-1}$ WSC is lost compared to hay samples during the conventional fermentation process [7]. In addition, due to the influence of oat-growing areas, most of them have lower WSC, which is not conducive to silage fermentation. Cellulases (endo-1,4-β-D-glucanases) are glycoside hydrolases that can hydrolyze plant cell walls, and they are produced by fungi, bacteria, and protozoans. They catalyze the hydrolysis of cellulose to monosaccharides or oligosaccharides products that can be used directly by bacteria [9]. Xylanases (endo-1,4-β-xylanase) catalyze the hydrolysis of xylan. The commercial source of xylanase is mainly produced by filamentous fungi [10].

In recent years, exogenous fibrolytic enzymes were used in silage fermentation to degrade the rate of fibers and increase the amount of WSC [11]. Meanwhile, the enzymes are used in ruminant feeds to increase the efficiency of feed degrading the cell wall of the forage [12]. The quality of forage silage was significantly improved by treatment with xylanases and cellulases, and this will also improve the digestibility of feed in the rumen [13]. Recent studies suggest that cellulase and *Lactobacillus plantarum* can improve the quality of hybrid *Pennisetum* silage. Cellulase has better nutrition preservation and ability to inhibit protein hydrolysis than *Lactobacillus plantarum* and sucrose [14]. Cellulose was hydrolyzed into glucose with cellulase treatment, which can be used directly by lactic acid bacteria [14]. Cellulose increased the content of lactic acid and decreased pH, acetic acid, and ammonia nitrogen content, and the abundances of *Lactobacillus* also increased significantly compared to no additive treatment; meanwhile, the cellulose treatment improved aerobic stability [15]. The addition of cellulase significantly modifies the in vitro digestibility of silage [16]. Xylanases are also used in silage and animal feed. Xylanases convert the xylan of the cell wall to sugars and can also provide microbial fermentation substrates; pretreatment of agricultural silage with xylanases improved nutritional value by reducing the content of hemicellulose in cell wall [13]. Xylanase-treated silage produces xylose, which was fermented to lactic acid by *Lactobacillus* to preserve silage [17]. In this study, we used oat with low water-soluble carbohydrates as experimental material and investigated whether cellulase and xylanase can reduce the fiber content in oat to provide fermentable sugars for microorganisms and promote silage fermentation.

## 2. Materials and Methods

### 2.1. Silage Preparation and Treatments

We derived the oat material from ArKhorchin Banner (43°77′ E, 120°78′ N), Chifeng, China; the oat was harvested during the heading period (20 June) and dried in the field for four hours. The dry matter content was 288.98 g/kg DM before ensiling, and the whole plant oat was chopped to 2–3 cm in length; then, all chopped oat materials were mixed and randomly packed into polyethylene bags (25 × 35 cm), and four samples were randomly selected for chemical and microbial composition analysis before ensiling. The treatment method was as follows: (i) control (CK, distributed water); (ii) silages

inoculated with xylanase (X); and (iii) silages inoculated with cellulase (C), which were provided by Shanghai Macklin Blochemical Co. Ltd., with xylanase and cellulase activities of 100,000 μ/g and 10,000 μ/g, respectively. An additional amount of two enzymes was 1000 μ/kg FW, which was sprayed fairly on the material of the oat. The control group (CK) was sprayed with an equal amount of distilled water, and each group was mixed and placed in a polyethylene bag with 300 g each; each group had 4 replicates. After silage, the samples were stored at room temperature (25 °C ± 2) away from light. Samples were opened for analysis after ensiling for 3 days (early stage of silage) and 60 days (late stage of silage) and after 5 days of aerobic exposure of oat silage. These samples were randomly taken for chemical composition and microbial composition analysis, and an aerobic stability study was conducted.

### 2.2. Chemical Composition and Fermentation Composition

All samples of oat were dried at 65 °C for 48 h to a constant weight to determine the dry matter (DM) content [11]. Subsequently, all dried samples were ground for chemical composition analysis. The WSC content was analyzed by colorimetry after reaction with anthrone reagent [18]. The FOSS Kjeltec 8400 was used to measure CP [19]. The ANKOM fiber analyzer (model: A2000 i; Beijing Anke Borui Technology Co., Ltd., Beijing, China) was used to determine ADF and NDF by the method of Van Soest et al. [20]. EE was measured using an ANKOM fat analyzer (model: XT15 i; Beijing Anke Borui Technology Co., Ltd., Beijing, China) [21]. Crude ash content was determined by a muffle furnace (model: SX2-10-12 N; Shanghai Yiheng Technology Co., Ltd., Shanghai, China) [22].

The liquid extract of silage was obtained by 10 g of sample mixed with 90 mL of distilled water, tapping for 2 min with a sterile homogenizer, and filtering it through four layers of coarse cotton cloth and filter paper. After the liquid extract of silage pH was determined immediately by a pH meter (model: LEICI pH S-3 C, Shanghai Yitian Scientific Instrument Co., Ltd., Shanghai, China). The liquid extract of silage organic acid was determined by high-performance liquid chromatography (model: Waters E2695, Milford, MA, USA), and the following chromatographic column was used for the separation of organic acids: RSpak series: KC-811 column 8 mm × 300 mm; mobile phase: 3 mmol mobile phase superior pure perchloric acid; flow rate: 1 mL/min; column temperature: 40 °C; detection wavelength: 210 nm; and injection volume: 5 μL. The content of ammonia nitrogen ($NH_3$-N) was measured by the phenol–hypochlorous acid colorimetric method of Broderick and Kang [23].

### 2.3. Microbial Sequencing

All samples to be tested for microbial DNA were extracted using the E.Z.N.A. stool DNA Kit (Omega Biotek, Norcross, GA, USA). The 16 S rDNA V3-V4 region of the Eukaryotic ribosomal RNA gene was amplified by PCR using the primers 341 F: CCTACGGGNG-GCWGCAG and 806 R: GGACTACHVGGGTATCTAAT (where the barcode is an eight-base sequence unique to each sample). PCR assays were performed at 95 °C for 2 min; followed by 27 cycles at 98 °C for 10 s, 62 °C for 30 s, and 68 °C for 30 s; and a final extension at 68 °C for 10 min. Amplicons were extracted from 2% agarose gels, purified using the AxyPrep DNA Gel Extraction Kit (Axygen Biosciences, Union City, CA, USA) according to the manufacturer's instructions, and quantified using QuantiFluor-ST (Promega, Madison, WI, USA). Purified amplicons were pooled in equimolar amounts, and paired-end sequences (2 × 250) were obtained with an Illumina HiSeq 2500 system (Illumina, San Diego, CA, USA) at Gene Denovo Biological Technology Co., Ltd. (Guangzhou, China).

### 2.4. Aerobic Stability

After 60 days of ensiling, the oat silage was opened and placed in a 1 L sterile plastic bottle. A multichannel temperature recorder (model: MDL-1048 A; Shanghai Tianhe Automation Instrument Co., Ltd., Shanghai, China) was used to record temperature changes;

the data were recorded once every ten minutes. Aerobic stability is defined as the time required for a temperature increase of 2 °C in silage [8].

### 2.5. Statistical Analysis

Analyses were performed with Statistical Packages for the Social Sciences (SPSS, Version 26.0). One-way ANOVA was performed on the chemical composition data of oat silage. During the additives period, silage period, and under the fixed influence of additives ×silage period, the chemical composition and fermentation quality data of silage were analyzed by two-way ANOVA. We used Duncan's multiple comparisons to determine statistical differences between mean values and considered them to be significant at $p < 0.05$. Bioinformatic analysis was performed using Omicsmart, a dynamic real-time interactive online platform for data analysis (http://www.omicsmart.com (accessed on 20 June 2023)). We used Blast for sequence alignment, the SILVA database to annotate the characteristic sequence of each representative sequence, and QIIME software to analyze the alpha diversity index. The Spearman correlation analysis was completed with R.

## 3. Results

### 3.1. Characteristics of Oat before Ensiling

The chemical composition of oat before ensiling is shown in Table 1; the pH was 6.41, the content of DM was 290.24 g/kg FW, the content of NDF, ADF, acid detergent lignin (ADL), CP, WSC, Ash, and EE were 663.04, 378.19, 56.80, 123.47, 62.13, 62.19, and 15.50 g/kg DM, respectively, the content of lactic acid bacteria (LAB) was 1.51 log CFU per g FW, and the content of yeast was 2.01 log CFU per g FW.

**Table 1.** Chemical composition of oat before ensiling.

| Items | Oat Silage | SEM |
|---|---|---|
| pH | 6.41 | 0.014 |
| DM (g/kg FW) | 290.24 | 0.218 |
| NDF (g/kg DM) | 663.04 | 0.455 |
| ADF (g/kg DM) | 378.19 | 0.325 |
| ADL (g/kg DM) | 56.80 | 0.149 |
| CP (g/kg DM) | 123.47 | 0.068 |
| WSC (g/kg DM) | 62.13 | 0.087 |
| Ash (g/kg DM) | 62.19 | 0.056 |
| EE (g/kg DM) | 15.50 | 0.145 |
| LAB (lg CFU/g) | 1.51 | 0.006 |
| Yeast (lg CFU/g) | 2.01 | 0.038 |

FW, fresh weight; DM, dry matter; NDF, neutral detergent fiber; ADF, acid detergent fiber; ADL, acid detergent lignin; CP, crude protein; WSC, water-soluble carbohydrates; EE, ether extract; LAB, lactic acid bacteria; SEM, standard error of the mean.

### 3.2. Nutrient Composition of Oat during Different Treatments and Periods of Ensiling

Additives and silage time have different effects on the nutritional quality of oat silage (Table 2). The DM content of oat showed a decreasing trend with silage time. In the early stage of silage (3 days), the DM content of xylanase treatment significantly decreased compared to the control ($p < 0.05$). In the late stage of silage (60 days), the DM content of cellulase and xylanase treatments was markedly lower than the CK treatment. Different additives treatments and silage time had a significant impact on the NDF content of oat ($p < 0.01$). Compared with the CK treatment, both cellulase and xylanase treatments reduced the DNF content significantly ($p < 0.01$). The NDF content of cellulase and xylanase treatments in the early (3 days) and late stages of silage (60 days) was significantly ($p < 0.01$) lower than that of the CK treatment; during the aerobic exposure period, the lowest NDF content was 545.79 g/kg DM treated with cellulase. The ADF content of cellulase and xylanase treatments markedly decreased in the late stage of silage (60 days) and during the aerobic exposure period and gradually decreased with the increase of silage time.

The xylanase treatment markedly reduced the ADL content during the aerobic exposure period ($p < 0.05$), but different additive treatments did not show significant differences in ADL content during the early (3 days) and late stages of silage (60 days). Cellulase and xylanase treatments significantly reduced the cellulose content during the late stage of silage (60 days) and the aerobic exposure period, but not during the early stage of silage (3 days). Compared with the early stage of silage (3 days), under the same additive treatment, both the late stage of silage and the aerobic exposure period increased the CP content remarkably ($p < 0.01$), while the xylanase treatment markedly reduced the CP content in the early stage of silage (3 days, $p < 0.05$). Although the xylanase treatment increased the WSC content in the later stage of silage (60 days), it significantly decreased during the aerobic exposure period. The cellulase treatment markedly increased the WSC content during the aerobic exposure period. The additives treatment did not have a significant effect on the EE content of oat silage. Under the same treatment, the EE content was striking increased in the later stage of silage (60 days) compared to the early stage of silage (3 days) and the aerobic exposure period ($p < 0.01$).

**Table 2.** Fermentation quality of oat silage during different treatments and periods of ensiling.

| Items [1] | Treatment [2] | Ensilage Period [3] | | | Mean [4] | SEM [5] | Significance [6] | | |
|---|---|---|---|---|---|---|---|---|---|
| | | 3 | 60 | AE | | | T | D | T × D |
| DM (g/kg FW) | CK | 297.05 [bA] | 307.77 [aA] | 295.75 [b] | 300.19 [A] | 0.620 | | | |
| | X | 287.02 [bB] | 297.25 [aB] | 296.43 [a] | 293.57 [B] | 0.120 | ns | ns | ns |
| | C | 295.11 [A] | 294.12 [B] | 292.62 | 293.95 [B] | 0.609 | | | |
| NDF (g/kg DM) | CK | 638.54 [aA] | 607.34 [cA] | 627.40 [bA] | 624.43 [A] | 0.172 | | | |
| | X | 626.70 [aB] | 568.63 [bB] | 576.15 [bB] | 590.49 [B] | 0.870 | ** | ** | ** |
| | C | 602.49 [aC] | 550.77 [bB] | 545.79 [bB] | 566.35 [C] | 0.429 | | | |
| ADF (g/kg DM) | CK | 329.44 [bB] | 346.58 [aA] | 357.01 [aA] | 344.34 [A] | 0.329 | | | |
| | X | 344.90 [aA] | 321.35 [bB] | 305.30 [bB] | 323.85 [B] | 0.301 | ** | ** | ** |
| | C | 339.31 [aB] | 308.22 [bB] | 315.48 [bB] | 321.00 [B] | 0.514 | | | |
| ADL (g/kg DM) | CK | 46.53 | 42.95 | 49.89 [A] | 46.46 | 0.297 | | | |
| | X | 50.46 [a] | 45.72 [b] | 36.95 [cB] | 44.38 | 0.329 | ns | ns | * |
| | C | 51.64 [a] | 50.66 [a] | 44.85 [bA] | 49.05 | 0.086 | | | |
| Hemicellulose (g/kg DM) | CK | 309.10 [aA] | 260.76 [bA] | 270.39 [bA] | 280.08 [A] | 0.379 | | | |
| | X | 281.80 [aB] | 247.28 [bB] | 240.85 [bB] | 266.64 [B] | 0.834 | ** | ** | ns |
| | C | 263.18 [aB] | 242.56 [bB] | 230.31 [bB] | 245.35 [C] | 0.690 | | | |
| Cellulose (g/kg DM) | CK | 282.91 [b] | 303.63 [aA] | 307.11 [aA] | 297.88 [A] | 0.397 | | | |
| | X | 294.44 [a] | 275.63 [bB] | 268.35 [bB] | 279.47 [B] | 0.435 | ns | ** | ** |
| | C | 287.67 [a] | 257.56 [bB] | 276.97 [bB] | 274.07 [B] | 0.504 | | | |
| CP (g/kg DM) | CK | 133.87 [bA] | 143.04 [a] | 143.65 [a] | 140.23 | 0.011 | | | |
| | X | 124.51 [bB] | 144.68 [a] | 145.19 [a] | 138.25 | 0.012 | ns | ** | ** |
| | C | 134.35 [bA] | 142.72 [a] | 142.10 [a] | 139.87 | 0.012 | | | |
| WSC (g/kg DM) | CK | 31.52 [a] | 17.89 [bB] | 16.19 [cb] | 21.87 | 0.054 | | | |
| | X | 30.80 [a] | 21.90 [bA] | 16.66 [c] | 23.12 | 0.097 | ns | ** | ** |
| | C | 33.08 [a] | 16.58 [bB] | 17.09 [b] | 22.25 | 0.075 | | | |
| EE (g/kg DM) | CK | 18.26 [b] | 27.23 [a] | 23.85 [b] | 23.11 | 0.129 | | | |
| | X | 17.98 [b] | 28.02 [a] | 24.48 [b] | 23.49 | 0.143 | ** | ns | ns |
| | C | 19.95 [b] | 27.76 [a] | 21.81 [b] | 23.17 | 0.159 | | | |

Values with different lowercase letters show significant differences among ensiling days in the same treatment; values with different capital letters show significant differences among treatments in the same ensiling day ($p < 0.05$); ns, not significant; * $p < 0.05$; ** $p < 0.01$. [1] NDF, neutral detergent fiber; ADF, acid detergent fiber; ADL, acid detergent lignin; DM, dry matter; CP, crude protein; WSC, water-soluble carbohydrates; EE, ether extract. [2] CK, control, no additive; X, silages inoculated with xylanase; C, silages inoculated with cellulase; [3] 3, after 3 days of silage; 60, after 60 days of silage; AE, after 5 days of aerobic exposure of silage. [4] The average values of different silage periods with the same additive. [5] SEM, standard error of means. [6] T, treatment; D, ensilage time; T × D, the interaction between treatment and ensilage time.

### 3.3. Dynamics of Temperature Fermentation Quality of Oat during Different Treatments and Periods of Ensiling

The aerobic stability levels of the cellulase and xylanase treatment groups were better than that of the CK group (Figure 1). Cellulase and xylanase treatments significantly reduced the pH during the late stage of silage (60 days) and the aerobic exposure period, but not during the early stage of silage (3 days, Table 3). The lowest pH of oat in the late stage of silage (60 days) with cellulase treatment was 4.39, which is significantly lower than other treatments ($p < 0.01$). The lactic acid content significantly increased with increasing silage time ($p < 0.01$). The additives treatment significantly increased the lactic acid content of oat silage in the early stage of silage, while cellulase treatment significantly increased the lactic acid content in the late stage of silage and aerobic exposure period compared to CK and xylanase treatment. Under the same additive treatment, the acetic acid content in the late stage of silage and aerobic exposure period was significantly higher than that in the early stage of silage. During the same silage period, both cellulase and xylanase treatments significantly increased the acetic acid content in the silage of the oat. Different additives and silage time treatments did not have a significant impact on propionic acid and butyric acid. The $NH_3$-N content in oat silage notably increased with the increase of silage time ($p < 0.01$). During the same silage period, both cellulase and xylanase treatments markedly increased the $NH_3$-N content in oat silage ($p < 0.01$).

**Table 3.** Fermentation quality of oat silage during different treatments and periods of ensiling.

| Items | Treatment [1] | Ensilage Period [2] | | | Mean [3] | SEM [4] | Signifificance [5] | | |
|---|---|---|---|---|---|---|---|---|---|
| | | 3 | 60 | AE | | | T | D | T × D |
| pH | CK | 4.97 [a] | 4.63 [Ab] | 4.91 [Aa] | 4.84 [A] | 0.058 | | | |
| | X | 4.85 [a] | 4.46 [Bb] | 4.65 [Bb] | 4.65 [B] | 0.095 | ** | ** | ns |
| | C | 4.94 [a] | 4.39 [Bc] | 4.59 [Bb] | 4.64 [B] | 0.043 | | | |
| Lactic acid (g/kg DM) | CK | 7.02 [B] | 24.36 [aC] | 22.32 [bC] | 17.90 [B] | 0.397 | | | |
| | X | 8.80 [bA] | 31.90 [aB] | 32.08 [aB] | 24.26 [A] | 0.320 | ** | ** | ** |
| | C | 9.05 [cA] | 35.98 [bA] | 40.25 [aA] | 28.43 [A] | 0.325 | | | |
| Acetic acid (g/kg DM) | CK | 1.29 [bA] | 1.43 [aA] | 1.53 [aA] | 1.42 | 0.028 | | | |
| | X | 0.88 [bB] | 1.36 [aB] | 1.41 [aB] | 1.22 | 0.050 | ** | ** | ** |
| | C | 0.88 [bB] | 1.22 [aB] | 1.35 [aB] | 1.15 | 0.021 | | | |
| Propionic acid (g/kg DM) | CK | 2.92 | 3.19 | 3.35 | 3.15 | 0.320 | | | |
| | X | 2.60 | 2.51 | 2.60 | 2.57 | 0.037 | ** | ns | ns |
| | C | 2.49 | 2.48 | 2.46 | 2.48 | 0.097 | | | |
| Butyric acid (g/kg DM) | CK | 2.92 | 3.19 | 3.35 | 3.15 | 0.320 | | | |
| | X | 2.60 | 2.51 | 2.60 | 2.57 | 0.037 | ns | ** | ns |
| | C | 2.49 | 2.48 | 2.46 | 2.48 | 0.097 | | | |
| $NH_3$-N (g/kg DM) | CK | 19.04 [Bc] | 39.45 [Bb] | 43.75 [Ca] | 34.08 [B] | 0.397 | | | |
| | X | 21.70 [Ac] | 47.25 [Ab] | 51.97 [Ba] | 40.31 [A] | 0.766 | ** | ** | ** |
| | C | 23.23 [Ac] | 51.08 [Ab] | 56.14 [Aa] | 43.48 [A] | 0.528 | | | |

Values with different lowercase letters show significant differences among ensiling days in the same treatment; values with different capital letters show significant differences among treatments in the same ensiling day ($p < 0.05$); ns, not significant; ** $p < 0.01$. [1] CK, control, no additive; X, silages inoculated with xylanase; C, silages inoculated with cellulase. [2] 3, after 3 days of silage; 60, after 60 days of silage; AE, after 5 days of aerobic exposure of silage. [3] The average values of different silage periods with the same additive. [4] SEM, standard error of means. [5] T, treatment; D, ensilage time; T × D, the interaction between treatment and ensilage time.

### 3.4. Microbial Community of Oat Silage during Different Treatments and Periods of Ensiling

Oat silage was mainly regulated by Firmicutes and Proteobacteria during different treatments and periods of ensiling, and Firmicutes had the highest relative abundance (Figure 2). Cellulase and xylanase treatments increased the abundance of Firmicutes and decreased the abundance of Proteobacteria in the early and late stages of silage, but the abundance of Firmicutes decreased and Proteobacteria increased significantly during aerobic exposure. Different ensiling periods contained different genus abundance structures, the dominant genera were *Weissella*, *Enterobacter*, and *Leuconostoc* in the early

stage of silage. During ensiling periods of late stage and aerobic exposure, the dominant genera were *Lactobacillus* and *Pediococcus*. In the late stage of silage, xylanase treatment markedly increased the abundance of *Pediococcus*, cellulase notably increased Lactobacillus ($p < 0.05$), and both cellulase and xylanase decreased the abundance of *Weissella* and *Enterococcus*. However, *Lactobacillus* abundance remarkably increased without additives during aerobic exposure ($p < 0.05$).

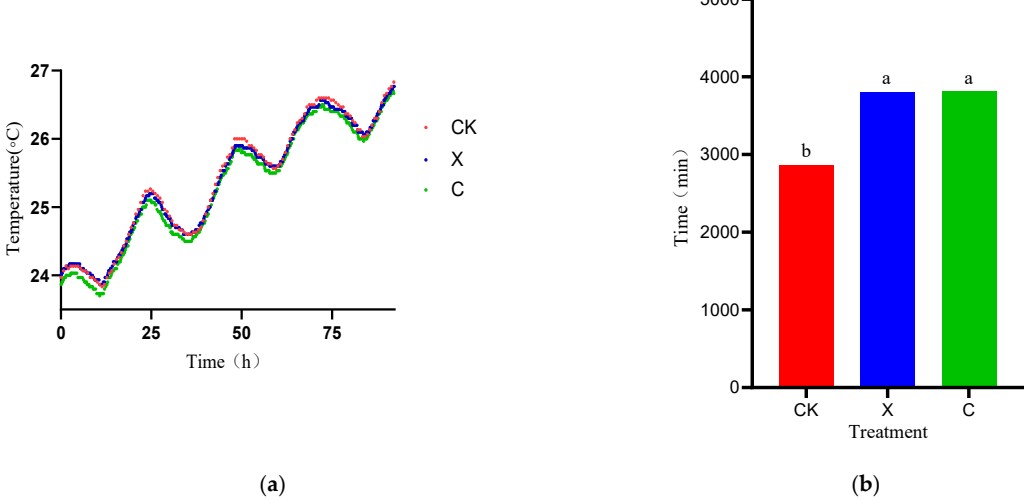

(**a**)                    (**b**)

**Figure 1.** Dynamics of oat silage temperature (°C) during aerobic exposure (**a**) and the time when silage temperature was 2 °C below initial temperature (**b**). Means with different lowercase letters (a, b) indicate significant differences in aerobic stability time ($p < 0.05$). CK, control, no additive; X, silages inoculated with xylanase; C, silages inoculated with cellulase.

(**a**)

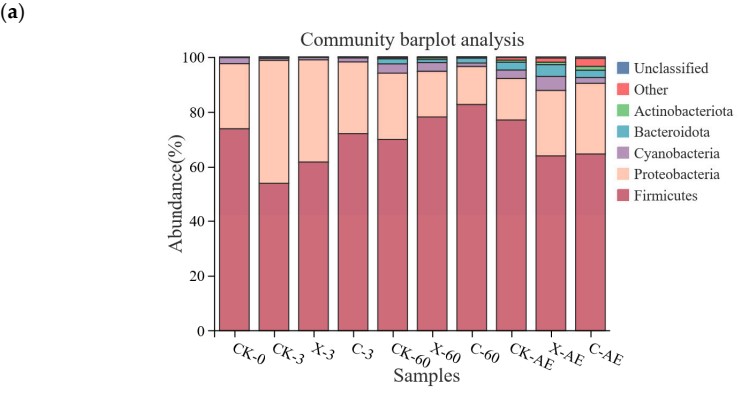

(**b**)

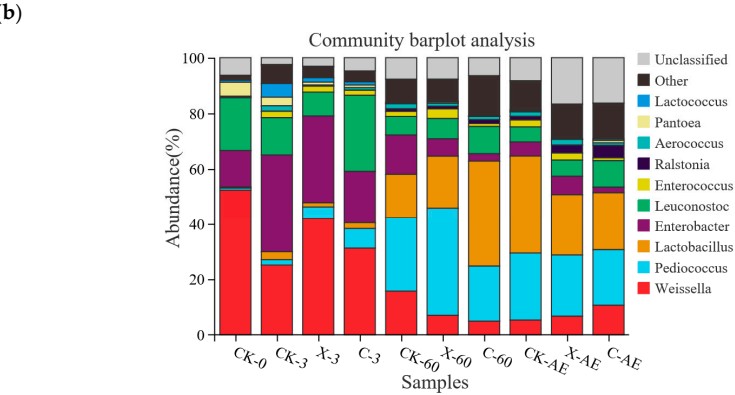

**Figure 2.** *Cont.*

(c)

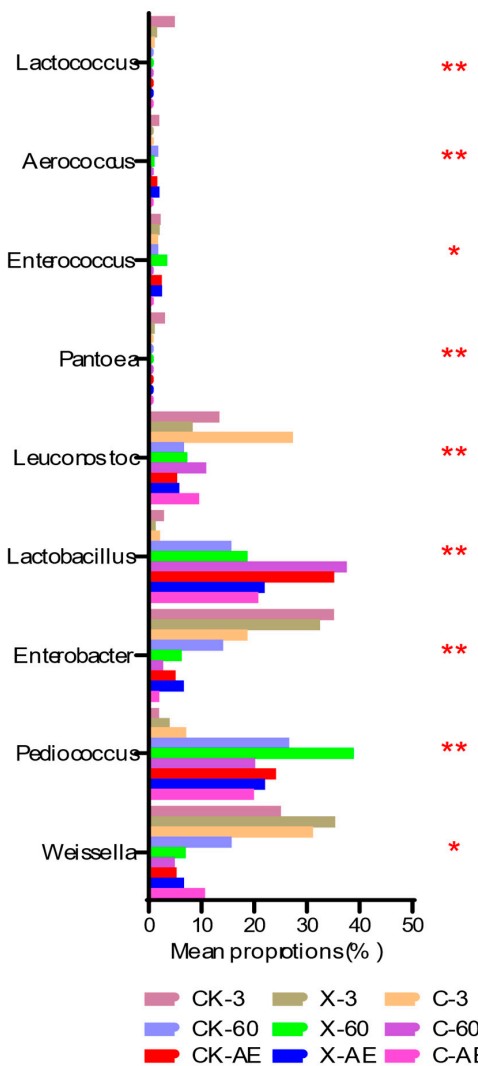

**Figure 2.** Dynamic changes in the relative abundance of bacteria at the phylum level (**a**) and genus level (**b**) during different treatments and periods of ensiling and the differences in genus level (**c**). * Significance at $p < 0.05$; ** significance at $p < 0.01$. CK-0, oat before ensiling; CK-3, no additive after 3 days of silage; X-3, inoculated with xylanase after 3 days of silage; C-3, inoculated with cellulase after 3 days of silage; CK-60, no additive after 60 days of silage; X-60, inoculated with xylanase after 60 days of silage; C-60, inoculated with cellulase after 60 days of silage; CK-AE, no additive after 5 days of aerobic exposure of silage; X-AE, inoculated with xylanase after 5 days of aerobic exposure of silage; C-AE, inoculated with cellulase after 5 days of aerobic exposure of silage.

Figure 3 shows the α-diversity of the bacterial community of silages. Comparing the early and late stages of oat silage, the Sob index and Chao index increased remarkably during aerobic exposure ($p < 0.05$). There was no significant difference in the Sob index between the early and late stages of silage, but there was a marked difference in the Chao index between the early and late stages of silage under xylanase treatment ($p < 0.05$). The Shannon index and Simpson index have a similar changing law during different treatments and periods of ensiling. The Shannon index and Simpson index of cellulase and xylanase treatments during aerobic exposure were significantly ($p < 0.05$) higher than during the early stage of silage. The Shannon index and Simpson index showed the lowest level during the early stage of silage ($p < 0.05$).

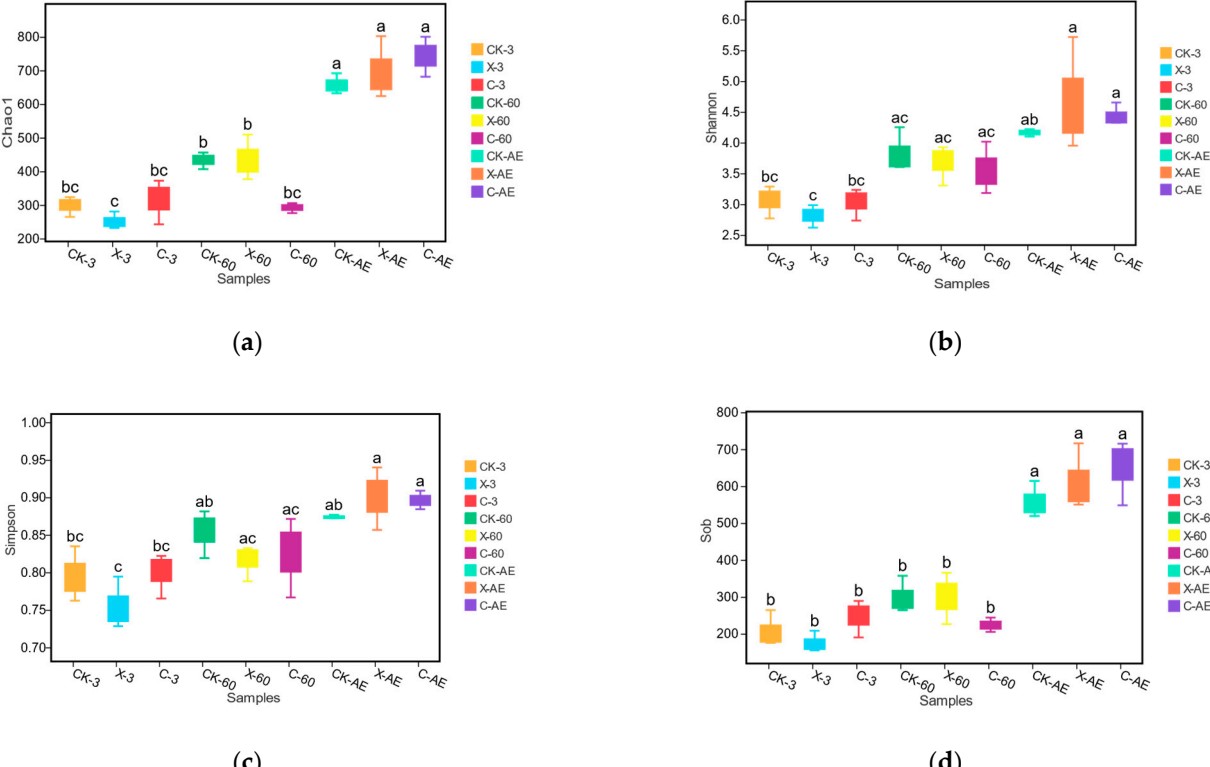

**Figure 3.** Differences in bacterial (**a–d**) community diversity and richness in oat silage during different treatments and periods of ensiling, different lowercase letters indicate significant differences between samples. (**a**) Sob index; (**b**) Chao index; (**c**) Shannon index; (**d**) Simpson index. CK-3, no additive after 3 days of silage; X-3, inoculated with xylanase after 3 days of silage; C-3, inoculated with cellulase after 3 days of silage; CK-60, no additive after 60 days of silage; X-60, inoculated with xylanase after 60 days of silage; C-60, inoculated with cellulase after 60 days of silage; CK-AE, no additive after 5 days of aerobic exposure of silage; X-AE, inoculated with xylanase after 5 days of aerobic exposure of silage; C-AE, inoculated with cellulase after 5 days of aerobic exposure of silage.

The PCoA shown in Figure 4a examines the correlations among the community structures of the silage bacterial community. The cellulase and xylanase treatments had a clear separation and difference of bacterial communities at different silage periods, suggesting that the bacterial composition was changed with different additives treatments and periods of ensiling. Different grouping factors have high explanatory power for the differences in samples, and the reliability of the test is high. ($R^2$ = 0.8072, $p$ = 0.001; Figure 4b). The Venn diagram of bacteria shown in Figure 4c revealed 54 shared OTUs among the treated bacteria samples; there were more OTUs unique to the xylanase treatment groups at the later stage of ensiling and aerobic exposure than in other treatments, and the lowest OTUs number was found in the X-3 group (65).

The functional abundance heat map is shown in Figure 5; there was a markedly negative correlation between different treatments and the metabolic function of oat before ensiling and during the early stage of silage ($p < 0.05$), while there was a significant positive correlation between different treatments and the metabolic function during the later stage of silage and aerobic exposure ($p < 0.05$), especially in the cellulase and xylanase treatment groups. The metabolic function in the early stage of silage was significantly ($p < 0.05$) lower than that in the later stage of silage and aerobic exposure period. Under different silage periods, cellulase and xylanase treatments did not have a significant impact on carbohydrate metabolism, metabolism of cofactors and vitamins, or metabolism of other amino acids. However, the cellulase treatment strikingly increased amino acid metabolism,

lipid metabolism, and metabolism of terpenoids and polyketides to the CK group during the late stage of ensiling ($p < 0.05$).

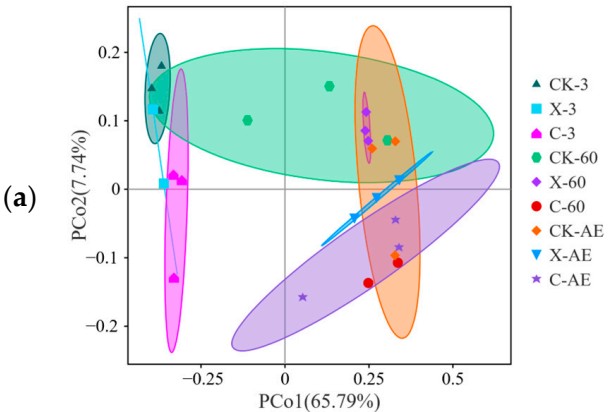

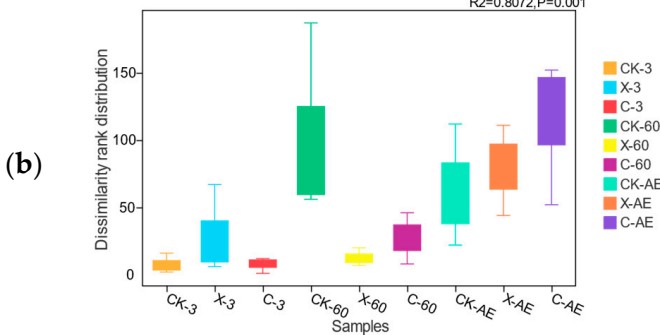

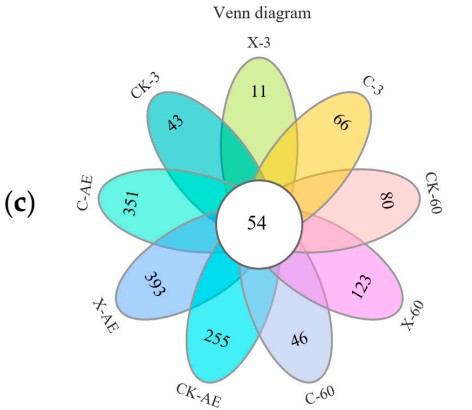

**Figure 4.** Principal component analysis of bacterial communities' (**a**) operational taxonomic units (OTUs) in oat silage during different treatments and periods of ensiling. Adonis test at the level of the distance of Bray–Curtis (**b**) in oat silage during different treatments and periods of ensiling. Venn diagram of bacterial (**c**) operational taxonomic units (OTUs) in oat silage during different treatments and periods of ensiling. CK-3, no additive after 3 days of silage; X-3, inoculated with xylanase after 3 days of silage; C-3, inoculated with cellulase after 3 days of silage; CK-60, no additive after 60 days of silage; X-60, inoculated with xylanase after 60 days of silage; C-60, inoculated with cellulase after 60 days of silage; CK-AE, no additive after 5 days of aerobic exposure of silage; X-AE, inoculated with xylanase after 5 days of aerobic exposure of silage; C-AE, inoculated with cellulase after 5 days of aerobic exposure of silage.

The Spearman analysis shown in Figure 6 clarifies the related relationships between silage parameters and bacterial species during different additives treatments and periods of ensiling. Silage pH was significantly ($p < 0.05$) negatively correlated with the abundances of *Sphingomonas* and *Anoxybacillus*. The content of LA was notably positively correlated with the abundance of *Ralstonia* and *Sphingomonas* ($p < 0.01$) but negatively correlated with the abundance of *Cosenzaea* and *Carnobacterium*. The content of AA was positively correlated with the abundance of *Lactobacillus_acidipiscis* ($p < 0.01$) and *Nocardioides* ($p < 0.05$). The CP content was strikingly positively correlated with the abundance of *Carnobacterium* ($p < 0.05$) but negatively correlated with *Sphingomonas* ($p < 0.01$). The WSC content was notably positively correlated with the abundance of *Cosenzaea* ($p < 0.01$) but negatively correlated with *Lactobacillus_acidipiscis*, *Ralstonia*, and *Anoxybacillus* ($p < 0.05$).

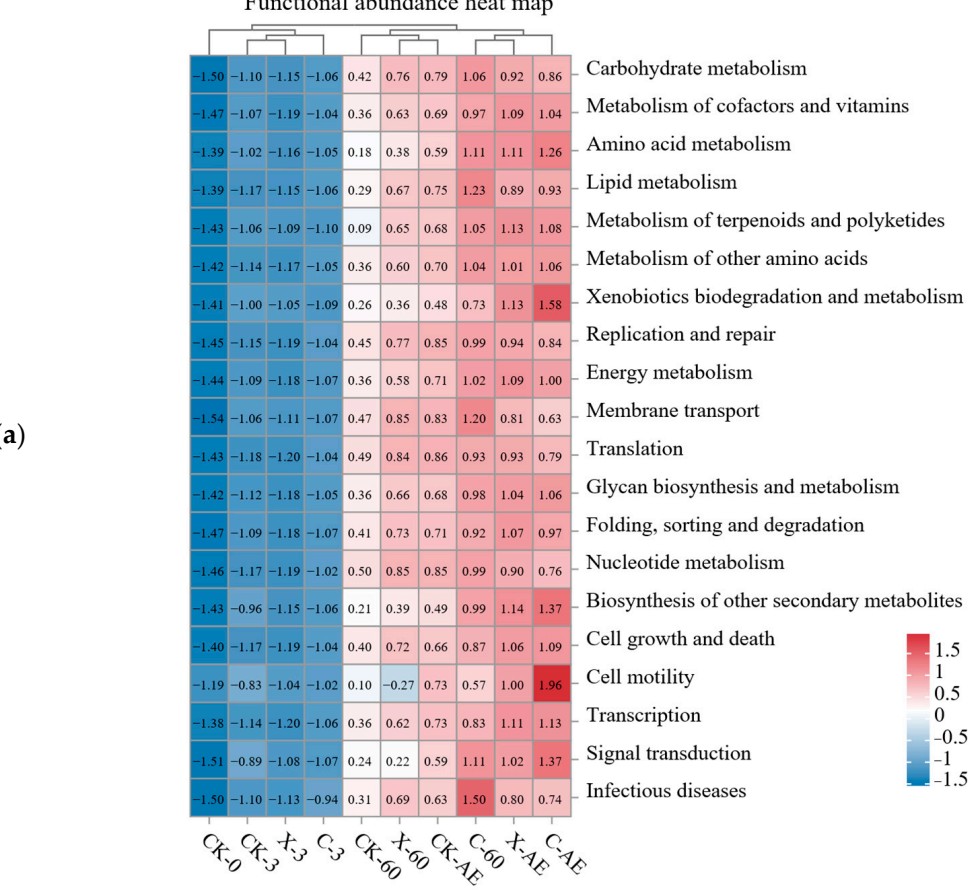

(**a**)

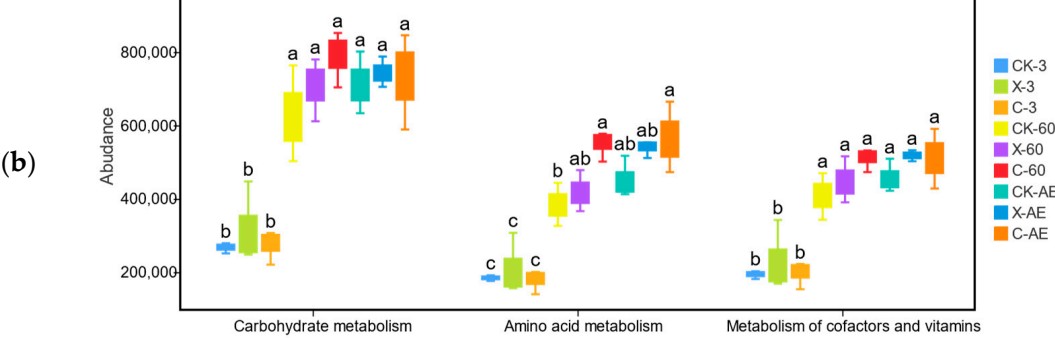

(**b**)

**Figure 5.** *Cont.*

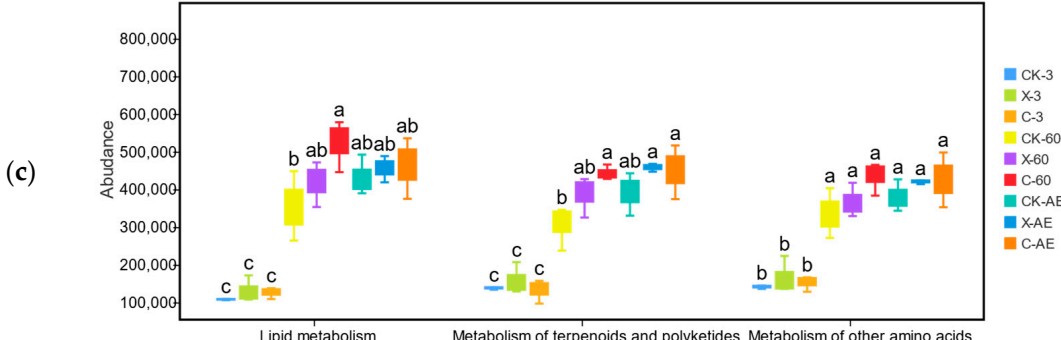

**Figure 5.** Functional abundance heat map (**a**) on classification level 2 in oat silage during different treatments and periods of ensiling. Metabolic differences analysis of carbohydrate, amino acid and cofactors and vitamins (**b**) in oat silage during different treatments and periods of ensiling different. Metabolic differences analysis of lipid, terpenoids and polyketides and other amino acids (**c**) in oat silage during different treatments and periods of ensiling. Different lowercase letters indicate significant differences between the same metabolic level.CK-3, no additive after 3 days of silage; X-3, inoculated with xylanase after 3 days of silage; C-3, inoculated with cellulase after 3 days of silage; CK-60, no additive after 60 days of silage; X-60, inoculated with xylanase after 60 days of silage; C-60, inoculated with cellulase after 60 days of silage; CK-AE, no additive after 5 days of aerobic exposure of silage; X-AE, inoculated with xylanase after 5 days of aerobic exposure of silage; C-AE, inoculated with cellulase after 5 days of aerobic exposure of silage.

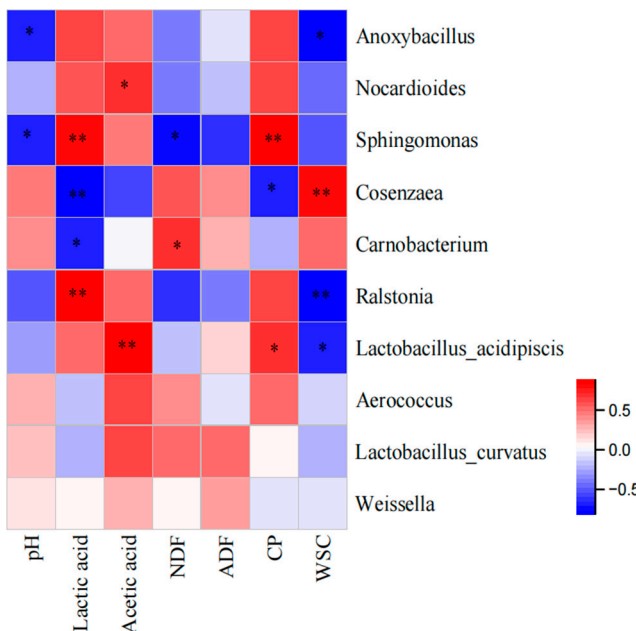

**Figure 6.** Spearman analysis between silage parameters and bacterial species during different treatments and periods of ensiling. * Significance at *p* < 0.05; ** significance at *p* < 0.01; NDF, neutral detergent fiber; ADF, acid detergent fiber; CP, crude protein; WSC, water-soluble carbohydrates.

## 4. Discussion

### 4.1. Material Characteristics and Silage Quality

The composition of the fresh forage in this experiment was within the expected range for oat. Compared to the previous reports, the DM, pH, WSC, NDF, and ADF of oat were higher, but yeast and LAB counts were lower [8,24]. Compared with the above-mentioned studies, Li H. et al. [24] found that the CP content was higher, but Wang S. et al. [8] found that the CP content was lower. The differences in the nutritional composition of oat may be

caused by cultivation, climatic conditions, soil fertility, and growth period. In this study, oat was harvested during the heading period, which may result in differences in nutritional composition. The WSC is an important factor affecting the fermentation quality. High-quality fermented silage needs sufficient WSC content (60–80 g/kg DM) for the growth of lactic acid bacteria [25]. In this study, the oat silage WSC was 62.19 g/kg, which met the minimum requirement, but the WSC content decreased rapidly during the silage process. If the fermentation substrate is insufficient, it will lead to fermentation failure and a decrease in nutritional quality [7]. Oat is rich in cellulose and hemicellulose (polysaccharide), which makes it possible to improve the fermentation quality of oat silage. A LAB number of more than 5.0 log10 cfu/g during ensiling achieves a good preservation [18]. In this study, the LAB content was 1.51 log CFU per g FW, which is very poor for the fermentation of oat silage, but each treated oat silage had a good fermentation quality, especially the cellulase and xylanase treatment groups. Therefore, it might be helpful to achieve a higher silage quality by reducing the fiber content, increasing the WSC content, and promoting fermentation by adding cellulase and xylanase.

The composition of nutrients is an important indicator for evaluating silage fermentation. The DM content was 290.24 g/kg FW, which is close to the optimal DM content range from 300 to 400 g $kg^{-1}$ FW for quality silage-making [26]. Cellulase and xylanase reduced the content of NDF and ADF in oat silage due to their ability to hydrolyze plant cell walls to produce fermentable sugars [27]. However, the ADF did not decrease with the xylanase treatment during the early stage of silage, possibly due to the time required for the action of xylanase on plant cell walls [28]. Similarly, there was a significant decrease in cellulose content during the late but not the early stage of silage under cellulase and xylanase treatments. The CP content significantly increased in the later stage of silage; this is different from previous studies that showed a decrease in CP content [29], which indicates that fermentation can also convert some non-protein nitrogen into amino acids and amino compounds; therefore, the protein content of silage will correspondingly increase [30]. Treatment with xylanase significantly increased the WSC content in the later stage of silage, indicating that xylanase can effectively degrade the xylan in plant cell walls and increase the fermentable sugar content in silage [17]. However, cellulase treatment did not increase the WSC content, which may be due to the fact that the pH in silage did not drop below 4.2, causing other harmful microorganisms to continue using sugar [14]. The ether extract significantly increased in the late stage of silage, which is due to fermentation that can be attributed to the breakdown of the cell walls of the forage material and the release of lipids from intracellular organelles and membranes; the microbes involved in the fermentation process also contribute to this increase as they help to break down the plant material and release more lipids [31].

Cellulase and xylanase treatments significantly reduced the silage pH in the later stage, which is consistent with previous research results [32]. The fibrotic enzyme promotes lactic acid fermentation by increasing the concentration of fermentable substrates, thus reducing the pH of silage [33]. The aerobic stability significantly increased under the treatment of cellulase and xylanase; this may be due to the increase in the content of fermentable sugars in oat silage, reducing the metabolic pressure of microorganisms, and thus improving the aerobic stability of the silage [33]. Adding cellulase and xylanase can quickly break down the plant cell walls, releasing more available nutrients and sugars; this can increase the growth rate of lactic acid bacteria and other beneficial microorganisms, thereby inhibiting the growth of acid-producing bacteria and reducing the acetic acid content, thereby significantly improving the quality of the fermented product [34]. The content of propionic acid and butyric acid in silage is influenced by various factors, including the type of raw materials, nutritional composition, harvesting time, and storage conditions [35]. However, in this study, there was no significant change in the content of propionic acid and butyric acid; this may be due to the suitable harvest period and environmental conditions for oat silage fermentation [36]. The ammonia nitrogen content increased significantly during the late stage of silage; this may be due to the fact that the microbial decomposition of

carbohydrates and proteins in silage increases the content of ammonia nitrogen in the silage [30].

*4.2. Microbial Community in Silage*

Silage is a fermentation process in which multiple microorganisms interact, and the composition of bacterial communities affects the quality of silage [37]. Whether before or after silage, Proteobacteria and Firmicutes dominated the entire bacterial community at the phylum level, which is consistent with the result of the previous study on red clover and napiergrass silage [38]. The diversity of the microbial community increased during the late stage of silage and the aerobic exposure period, especially the aerobic exposure period; this may be due to the pH not reaching the threshold to inhibit the growth of other microorganisms throughout the entire silage process, leading to an increase in bacterial diversity, but it may also improve the aerobic stability of silage [39]. Cellulase and xylanase treatments showed high levels of Proteobacteria at the phylum level during the early and late stages of silage, but significantly decreased during aerobic exposure, possibly due to the growth of other microorganisms during the aerobic exposure period [24]. Cellulase and xylanase have significant effects on the bacterial colony structure at the bacterial genus level during different silage periods, especially in the later stage of silage. Cellulase treatment significantly increased the abundance of *Lactobacillus* and reduced the pH, which is due to the decomposition of cellulose in the cell wall by cellulase, increasing the glucose content in the silage and providing sufficient substrate for *Lactobacillus* fermentation [40]. Xylanase treatment in the late stage of silage may also reduce the content of xylan in the cell wall and increase the content of xylose in the silage, leading to a significant increase in *Pediococcus* but not *Lactobacillus*, indicating that *Pediococcus* prefers to use xylan for fermentation, while *Lactobacillus* prefers to use glucose for fermentation [34]. Oat silage without addition during the aerobic exposure period has a high level of *Lactobacillus*. When the allowance of *Lactobacillus* is high, the number of aerobic bacteria in silage will also decrease, resulting in a lower ability to resist oxidation, leading to a decrease in the aerobic stability of silage [41].

Cellulase and xylanase treatments did not have a significant impact on the Sob, Chao, Shannon, and Simpson indexes, indicating that the additives treatment did not change the species richness of oat silage, which may be due to the absence of the addition and interference of exogenous bacteria [33]. The Sob index and Chao index in the aerobic exposure period were significantly higher than in the early and later stages of silage, indicating that the species diversity of samples was markedly increased due to environmental factors during aerobic exposure [37], and the Sob index and Chao index treated with cellulase and xylanase were slightly higher than the CK treatment. The Shannon and Simpson indexes showed that additives did not significantly improve the species diversity and evenness of oat silage bacteria. The Shannon and Simpson indexes were significantly higher in the aerobic exposure period than in the early stage of silage, which may be due to the high bacterial species diversity in the aerobic exposure period [24]. The principal component analysis clearly showed the differences between groups of bacterial communities, indicating that different treatment groups changed the differences in bacterial communities, especially during the aerobic exposure stage, where there were significant differences in bacterial colonies compared to the early and late stages of silage, which may be caused by environmental factors during the aerobic exposure period [37]. The abundance of bacteria and the number of special OTUs were higher during the aerobic exposure period, and the cellulase and xylanase treatment groups had more special OUT numbers, which may also contribute to the aerobic stability of silage [42]. Compared to the early stage of silage, the metabolic levels in the later stage of silage and aerobic exposure period were significantly increased. In the later stage of silage, cellulase treatment significantly increased the levels of amino acid metabolism, lipid metabolism, terpenoids metabolism, and polyketides metabolism compared to the CK treatment; this may be due to the increase in the content of fermentable substrates in bacteria with cellulase treatment, which reduced the environmental pressure for bacterial growth and thus increased the metabolic level [43].

## 5. Conclusions

The study indicated that cellulase and xylanase in different silage stages had significant effects on chemical compositions and fermentation quality, and cellulase treatment had better results although xylanase treatment obtained more WSC. Both cellulase and xylanase changed the abundance of the bacterial community, the aerobic exposure stage in particular, and they all improved the aerobic stability of oat silage. Although cellulase and xylanase reduced the NDF and ADF content, water-soluble carbohydrates were not well preserved with any treatment during the aerobic exposure stage, so how to keep more WSC in the oat silage is still a challenge.

**Author Contributions:** Methodology and software, M.L. and Z.W.; validation, formal analysis, investigation, resources, and data curation, W.L.; writing—original draft preparation, W.L. and Q.S.; writing review and editing, L.S. and S.D.; project administration, Y.J. and Z.W.; funding acquisition, Y.J. and G.G. All authors have read and agreed to the published version of the manuscript.

**Funding:** This research was co-financed by the Program for Technology Project of Inner Mongolia (2020GG0032) and National Dairy Technology Innovation Center (2021-National Dairy Innovation Center-1), China.

**Institutional Review Board Statement:** Not applicable.

**Informed Consent Statement:** Not applicable.

**Data Availability Statement:** The sequencing data for the 16 S rRNA gene sequence were stored at NCBI with BioProject accession number PRJNA980968.

**Conflicts of Interest:** The authors declare no conflict of interest.

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
