# Peer review of "Effects of Cellulase and Xylanase Addition on Fermentation Quality, Aerobic Stability, and Bacteria Composition of Low Water-Soluble Carbohydrates Oat Silage"

_fermentation, doi:10.3390/fermentation9070638_

Round 1

Reviewer 1 Report

The manuscript of Wei Liu and co-authors is devoted to the study of the influence of cellulase and xylanase enzymes on the quality of products during the fermentation of oat silage and on the composition of the microbial community.

In general, the manuscript corresponds to the subject of the journal Fermentation, the authors received a lot of data on the stated topic.

However, there are a number of remarks. Two main ones. 1. Sampling time. 2. English.

The authors provide data on the ensiling process based on sampling after three and 60 days of fermentation. In my opinion, there is not enough data to talk about dynamics (as Line 180). If the indicators are compared at two points - the beginning of fermentation and its end, then you can only record the changes obtained, then these data are not enough to track the dynamics of the process.

Other remarks. In the abstract, in order not to complicate the understanding of the results, it is necessary to first give the full names for all abbreviations. The same applies to the text of the article - the abbreviation must follow the full title.

Line 26 "increase really" cannot be used as a quantitative measure of the results obtained.

the entire text should be checked  for the presence of extra hyphens (line 74, 84, etc.) or the absence of spaces between words (lines 54, 122). All Latin names of strains, including those in the list of references, must be italicized.

I am not a native speaker, but it seems to me that the text requires careful checking of English, since a number of sentences do not make sense (lines 58-59; 366) and some sentences require the passive form of the predicate (line 69; 72, 79, 92, 113, 354, 361).

Line 80 - what processing are we talking about? Cellulose or cellulase?

Lines 81 and 82 - why is the word "silage" alternately capitalized and lowercase? What was the reason?

Lines 124-125. What was conditions for HPLC?

Section 2.3. The authors are requested to clarify the conditions of microbial profiling. Eukaryotes do not have 16S ribosomes, they are prokaryotes. You should specify the conditions for bioinformatics analysis for the given Shannon, Sob and Chao indices.

Results Table 1. What is NAF? (this is a typo and should be NDF?)? and where is the data for FW and NDF?

The description of the data (lines 178-182) for the ADF does not exactly match the data in the table. After the aerobic stage, the cellulase treatment variant shows an increase rather than a decrease in ADL. It's not entirely clear what the "Mean" column refers to. To all three indicators? What is the meaning of it?

Line 189: the text refers to an increase in WSC content, and the table provides data for the xylanase variant - a decrease in indicators (30-21-16).

Line 340. on WSC content. Table 1 gives a value of 63.13 and 62.19 in the text. What is the correct number?

I am not a native speaker, but it seems to me that the text requires careful checking of English, since a number of sentences do not make sense (lines 58-59; 366) and some sentences require the passive form of the predicate (line 69; 72, 79, 92, 113, 354, 361). The text as a whole is very difficult to read precisely because of strange grammatical constructions

Author Response

Dear Reviewer

Thank you for your letter and for the reviewers comments concerning our manuscript entitled “Effects of cellulase and xylanase addition on fermentation quality, aerobic stability, and bacteria composition of low water-soluble carbohydrates oat silage” (fermentation-2474348)。Those comments are all valuable and very helpful for revising and improving our paper, as well as the important guiding significance to our researches. We have studied comments carefully and have made correction which we hope meet with approval. Revised portion are marked in red in the paper. The main corrections in the paper and the responds to the reviewer's comments refer to the PDF file.

Reviewer 2 Report

I remain in doubt or probably not clear in the introduction section, the need or importance of this type of study.

In general, it is well written. Only a few details were observed in the text.

It is recommended that at least the first time an acronym appears, its meaning should be indicated, also in the results section; in tables or figures, sometimes it is difficult to remember the meaning through the text (NDF, ADF, CP, EE, WSC...).

Line 84, 85, error observed “Im-proves, hemi-cellulos”

Line 112; Specify or deeper, the conditions under which the method was carried out.

Table 1; NAF or NDF?

Table 1, 2, 3; report the letters "a or A", "b or B" in superscript.

Author Response

Dear Reviewer:

    Thank you for your letter and for the reviewers comments concerning our manuscript entitled “Effects of cellulase and xylanase addition on fermentation quality,aerobic stability, and bacteria composition of low water-soluble carbohydrates oat silage”(fermentation-2474348). Those comments are all valuable and very helpful for revisingand improving our paper, as well as the important guiding significance to our researches.We have studied comments carefully and have made correction which we hope meetwith approval. Revised portion are marked in red in the paper. The main corrections in the paper and the responds to the reviewer's comments refer to the PDF file.
